# Antiviral Effect of hBD-3 and LL-37 during Human Primary Keratinocyte Infection with West Nile Virus

**DOI:** 10.3390/v14071552

**Published:** 2022-07-15

**Authors:** Céline Chessa, Charles Bodet, Clément Jousselin, Andy Larivière, Alexia Damour, Julien Garnier, Nicolas Lévêque, Magali Garcia

**Affiliations:** 1Laboratoire de Virologie, CHU de Poitiers, 2, Rue de la Milétrie, CS 90577, 86000 Poitiers, France; celine.chessa@chu-poitiers.fr (C.C.); clement.jousselin@chu-poitiers.fr (C.J.); andy.lariviere@chu-poitiers.fr (A.L.); alexia.damour@u-bordeaux.fr (A.D.); nicolas.leveque@chu-poitiers.fr (N.L.); 2Laboratoire Inflammation, Tissus Epithéliaux et Cytokines, LITEC, UR 15560, Université de Poitiers, Pôle Biologie Santé, Bâtiment B36, 1, Rue Georges Bonnet, TSA 51106, CEDEX 09, 86073 Poitiers, France; charles.bodet@univ-poitiers.fr; 3Qima Bioalternatives, 1 Rue des Plantes, 86160 Gençay, France; julien.garnier@qima.com

**Keywords:** antimicrobial peptides, LL-37, hBD-3, West Nile virus, flavivirus, immunomodulation, antiviral, keratinocytes

## Abstract

West Nile virus (WNV) is an emerging flavivirus transmitted through mosquito bites and responsible for a wide range of clinical manifestations. Following their inoculation within the skin, flaviviruses replicate in keratinocytes of the epidermis, inducing an innate immune response including the production of antimicrobial peptides (AMPs). Among them, the cathelicidin LL-37 and the human beta-defensin (hBD)-3 are known for their antimicrobial and immunomodulatory properties. We assessed their role during WNV infection of human primary keratinocytes. LL-37 reduced the viral load in the supernatant of infected keratinocytes and of the titer of a viral inoculum incubated in the presence of the peptide, suggesting a direct antiviral effect of this AMP. Conversely, WNV replication was not inhibited by hBD-3. The two peptides then demonstrated immunomodulatory properties whether in the context of keratinocyte stimulation by poly(I:C) or infection by WNV, but not alone. This study demonstrates the immunostimulatory properties of these two skin AMPs at the initial site of WNV replication and the ability of LL-37 to directly inactivate West Nile viral infectious particles. The results provide new information on the multiple functions of these two peptides and underline the potential of AMPs as new antiviral strategies in the fight against flaviviral infections.

## 1. Introduction

Arboviruses are emerging single-stranded positive-sense RNA viruses, pathogenic for humans, transmitted by blood-sucking arthropods, predominantly mosquitoes. First described in Uganda in 1937 [1], the flavivirus West Nile virus (WNV) is an arbovirus affecting all continents and considered endemic in Europe, and the neighboring countries of the Mediterranean basin, with reports of cases each year and epidemics in some years [2,3,4]. The year 2018 was marked by a particularly large number of cases in Europe with 2083 infections reported, which represented more than all the cases declared over the previous seven years (1832 cases from 2011 to 2017) [5]. WNV can be responsible for a wide range of clinical manifestations from mild flu-like illness to encephalitis. There is, to date, no human vaccine or antiviral to combat WNV infections.

In humans and other mammals, WNV infection starts with inoculation of the virus during the blood meal of the mosquito, mainly into the extravascular compartment of the epidermis and the dermis, simultaneously with its antihemostatic, anticoagulant, anti-inflammatory and vasodilatory saliva [6,7]. Keratinocytes are the main cells in the epidermis. They have been demonstrated to be permissive, both in vivo and in vitro, to WNV replication, amplifying the viral load before the virus spreads through the body [8,9]. Conversely, keratinocytes also act as immune cells that can initiate an innate immune response to fight viral infection [9]. Indeed, they harbor highly conserved pathogen recognition receptors (PRRs) such as transmembrane toll-like receptors (TLRs) or cytosolic RIG-I-like receptors (RLRs) that contribute to the initiation of innate immune response [10]. After sensing single-stranded RNA genomes or double-stranded RNAs synthesized during the WNV replication cycle, PRRs are involved in the activation of signaling pathways leading to the induction of expression of chemokines, cytokines such as types I and III interferons (IFNs), and antimicrobial peptides (AMPs), which constitute the first line of defense of the host [11,12]. AMPs are key players in innate immunity. AMPs are 12 to 50 amino-acid peptides, mostly amphiphilic and cationic, exerting broad spectrum antimicrobial activity. The antiviral properties of AMPs may combine intracellular activity, on translation, viral RNA synthesis or virus particle assembly, with extracellular inactivation, by envelope disruption or aggregation of viral particles, to fight viruses [13,14,15,16,17,18,19]. Moreover, in addition to their direct antiviral activities on the virus replication cycle, AMPs display immunomodulatory properties that contribute to host defense, including chemotactic activity, proliferation and differentiation of immune cells, and regulation of cytokine/chemokine production [20,21]. In the skin, keratinocytes are the main cells involved in AMP production (reviewed in [22]). They produce and secrete at least nine AMPs including the human cathelicidin LL-37 and type 3 human β-defensin (hBD-3). LL-37 is a 37-amino acid residue, amphipathic, α-helical peptide and the only known human cathelicidin, while peptides of this large family have been isolated from numerous non-human species [23,24]. Defensins are small molecules of between 24 and 42 amino acids characterized by a β-sheet structure with 3 disulfide bounds. In humans, defensins are divided into α-defensins, referred to as human neutrophil peptides (hNPs), and β-defensins (hBDs) expressed in myeloid and epithelial cells. There are about 37 hBDs [21,25]. Four (hBD-1 to -4) have been detected in the epidermis. Expression of LL-37 and hBD-3 by keratinocytes has been shown to be rapidly induced following infection with viruses such as type 2 herpes simplex virus, vaccinia virus and the arboviruses, dengue virus (DENV) or WNV [21,26,27,28,29,30]. Furthermore, these two peptides display antiviral properties against many viruses that replicate in the skin, including DENV and Zika virus (ZIKV) [14,16,31,32,33,34,35]. Finally, LL-37 has been reported to exert indirect antiviral activity through immunomodulatory properties, by interfering with TLR signaling and attracting immune cells at the site of infection, promoting an inflammatory context favorable to pathogen eradication (reviewed in [22]). All in all, by their production at the initial replication site of arboviruses and their direct and/or indirect antiviral activities, these two peptides represent credible candidates as molecules active against a virus for which we have no treatment or vaccine for humans. In this study, we assessed the antiviral properties of the skin AMPs LL-37 and hBD-3 against WNV. We investigated their ability to inhibit WNV replication and infectious particle production in human primary keratinocytes. Then, we unraveled the mechanism of their antiviral activity by successively studying their ability to modulate the host’s immune response or to act by a direct, virucidal action on the viral particle. The results provide new information on the multiple functions of these two peptides and underline the potential of AMPs as new antiviral strategies in the fight against flaviviral infections.

## 2. Materials and Methods

### 2.1. Isolation and Culture of Normal Human Epidermal Keratinocytes from Human Skin Samples

The Ethics Committee of the Poitiers University Hospital approved the use of human skin samples for research studies. After the provision of fully informed consent, normal abdominal or breast skin was obtained from patients undergoing plastic surgery [36]. Normal human epidermal keratinocytes (NHEK) were seeded in sterile 24-well culture plates at a density of 2 × 10^4^ cells/well in keratinocyte serum-free medium (K-SFM; Gibco; Catalog No. 17005042) supplemented with bovine pituitary extract (BPE; Gibco, Waltham, MA, USA; Catalog No. 13028014) 25 µg/mL and epidermal growth factor (EGF; Gibco, Waltham, MA, USA; Catalog No. PHG0314) 0.25 ng/mL and cultured to 80% confluence. Cells were then starved overnight in K-SFM without growth factors before stimulation.

### 2.2. The Peptides and Their Cytotoxicity

Recombinant human LL-37 and hBD-3 were purchased from Invivogen (Toulouse, France; Catalog No. 154947-66-7) and PeproTech (Cranbury, NJ, USA; Catalog No. 500-P241), respectively. A 1 mg/mL stock solution of LL-37 and 500 µg/mL of hBD-3 was prepared by dissolving lyophilized synthetic peptides in sterile water, then aliquoted and frozen at −80 °C until use. LL-37 and hBD-3 cytotoxicity on NHEK was determined following incubation for 48 h with concentrations ranging from 1 to 20 µg/mL using the cell proliferation kit II (XTT, Roche Diagnostics, Meylan, France; Catalog No. 11465015001), as previously described [37]. For LDH assay, supernatants were collected after 24 h and mixed with 500 µL of PBS—0.1% Triton X-100 (Sigma-Aldrich, Saint-Quentin-Fallavier, France; Catalog No. 806,552 and 9036-19-5). Cells were lysed with 1 mL of PBS—0.1% Triton X-100 and sonicated for 30 s. The LDH released was measured with the Cobas (Roche) analyzer and viability was calculated, yielding the ratio between the LDH released in supernatants and the total LDH measured in both supernatants and cell lysates.

### 2.3. The Virus Strain and Its Production

The WNV strain used in this work belongs to lineage 1 and was isolated from a human brain during the epidemic that occurred in Tunisia in 1997 (kindly gifted by the French National Reference Center on Arboviruses, Marseille, France). The viral stock was produced on the *Aedes albopictus* clone C6/36 cells (ATCC CRL-1660). Insect cells were cultivated in Leibovitz’s L-15 medium (Gibco, Waltham, MA, USA; Catalog No. 11415049) supplemented with 2% of tryptose phosphate (Gibco, Waltham, MA, USA; Catalog No. 260200) and 5% of FBS in 75-cm^2^ tissue culture flask at 28 °C until 50% of confluency and then infected at a multiplicity of infection (MOI) of 0.01 for 72 h. Cell supernatants were clarified by centrifugation for 15 min at 1500× *g* and frozen at −80 °C in cryotubes containing 500 μL of Leibovitz’s L-15 medium supplemented with 0.5 M sucrose and 50 mM HEPES. The final titer of the viral suspension was 10^7.97^ TCID_50_ (median tissue culture infection dose) per mL as determined by plaque assays on Vero cell monolayers as described below.

### 2.4. Stimulation Protocol with Poly(I:C)

Human primary keratinocyte cultures (60–80% of confluency) were stimulated by addition to the cell culture medium of poly(I:C) (low molecular weight, Invivogen, Toulouse, France; Catalog No. tlrl-picw), a synthetic analog of double-stranded (ds) RNA that mimics ds replicative intermediates generated during flavivirus genome replication, at a final concentration of 0.1 µg/mL, in 1 mL cell culture medium, in presence or absence of LL-37 or hBD-3 at final concentrations of 0.1, 1 and 10 µg/mL, before being incubated at 37 °C in 5% CO_2_ in K-SFM medium [12]. Cell monolayers were collected after 3 h incubation in order to perform, by RT-qPCR, transcriptomic analysis of inflammatory marker expression as described below.

### 2.5. Infection Protocol with WNV

Human primary keratinocyte cultures (60–80% of confluency) were incubated for 1 h with LL-37 or hBD-3 at final concentrations of 1 and 10 µg/mL in 1 mL cell culture medium. The virus at a MOI of 0.1 was then added to the cell supernatant containing the peptide and left in contact with the cells for 24 or 48 h at 37 °C in 5% CO_2_ in K-SFM medium [12]. The viral load in the cell monolayer and the cell culture supernatant was measured by RT-qPCR at the end of the 24 h incubation and compared between treated and not treated cells. After 48 h incubation, cell monolayers and cell culture supernatants were used to monitor inflammatory marker expression at the protein level as described below.

### 2.6. RNA Isolation

Total RNA extraction from keratinocyte monolayers was performed using the Nucleospin RNA II kit according to the manufacturer’s instructions (Macherey-Nagel, Illkirch, France; Catalog No. 740955.50). RNA was eluted in 30 µL of RNAse-free H_2_O before being quantified using the Nanodrop 2000 spectrophotometer (Thermo Scientific). In addition, RNA extraction was performed from 200 µL of keratinocyte supernatants on the NucliSENS easyMAG (Biomérieux) according to the manufacturer’s instructions.

### 2.7. Monitoring of the Expression of Inflammatory Markers

Total RNA (1 µg) was reverse transcribed using SuperScript II Reverse Transcriptase kit (Invitrogen, Toulouse, France; Catalog No. 18064022) according to the manufacturer’s instructions. Quantitative real time (RT)-PCR was performed in 96-well plates using LightCycler-FastStart DNA Master^Plus^SYBR GREEN I kit (Roche, France; Catalog No. 03515869001) on LightCycler 480 (Roche Diagnostics). Reaction mixtures consisted of 1X DNA Master Mix (Applied Biosystems, Thermo Fischer Scientific, Waltham, MA, USA; Catalog No. 4305719), 1 μM forward and reverse primers designed using Primer 3 software and 12.5 ng of cDNA template in a total volume of 10 µL. PCR conditions were as follows: 5 min at 95 °C, 40 amplification cycles comprising 20 s at 95 °C, 15 s at 64 °C and 20 s at 72 °C. Samples were normalized with regard to two independent control housekeeping genes (glyceraldehyde-phospho-dehydrogenase and 28S rRNA gene) and reported according to the ΔΔCT method as RNA fold increase: 2^ΔΔCT^ = 2^ΔCT sample − ΔCT reference^.

### 2.8. Viral Quantification by RT-qPCR

Viral quantification in cell supernatants and keratinocyte monolayers was performed using a one-step real time RT-PCR assay from 5 μL of total RNA in 96-well plates on Applied Biosystems 7500 thermocyler. Reaction mixtures consisted of 12.5 μL of SuperScript III Platinum One-Step qRT-PCR Master Mix (Invitrogen, Toulouse, France; Catalog No. 11732020), 0.5 μL (0.2 μM) of forward (5′-3′ GTGCGGTCTACGATCAGTTT) and reverse primers (5′-3′ CACTAAGGTCCACACCATTCTC), and 0.25 μL (0.1 μM) of 5′FAM and 3′Dark Quencher probe (5′-3′ AATGTGGGAAGCAGTGAAGGACGA), 0.5 μL of SuperScript III reverse transcriptase (Invitrogen; Catalog No. 11732020) and DNA polymerase platinum Taq (Invitrogen; Catalog No. 11732020), 0.5 μL of RNaseOUT (Invitrogen; Catalog No. 10777019) and 5.25 μL of water. The calibration range was performed with a transcript produced using a plasmid containing WNV genome deleted from genes coding structural proteins provided by Dr P.W. Mason (Microbiology and Immunology department, Texas University, Galveston, TX, USA). The transcripts were diluted in order to obtain a calibration range allowing the quantification of viral load from 10^2^ to 10^7^ RNA copies/mL. Each standard of the calibration curve was added in duplicate to each analysis.

### 2.9. Viral Quantification by End-Point Dilution Assay

Vero cells were seeded in 96-well plates the day before titration at the rate of 4 × 10^3^ cells/well in DMEM (Gibco, Waltham, MA, USA; Catalog No. 11960044) supplemented with 2% FBS (Gibco; Catalog No. 26140087). The suspension was successively diluted from 10^−1^ to 10^−9^ in DMEM medium supplemented with 2% FBS. Then, 100 μL of each dilution were deposited in 6 wells of a row. A reading was performed after 96 h of incubation at 37 °C in an atmosphere containing 5% CO_2_. The wells in which the cells had a cytopathic effect were considered positive for viral infection. The titer of the viral suspension was then determined using Kärber’s method for assessing the TCID50.

### 2.10. Virucidal Property Assessment

The virucidal properties of LL-37 and hBD-3 were assessed by pre-incubating 1 × 10^5.3^ TCID50/mL of the WNV stock solution with 10 µg/mL of the AMP in a final volume of 0.1 mL. After 1 h of incubation at 37 °C in a humidified atmosphere with 5% CO_2_, the residual infectious titer of the viral suspension was then measured by viral titration using the end-point dilution assay on Vero cells as described above and compared to that of the untreated viral suspension.

### 2.11. Viperin (RSAD2) Quantification by Immunofluorescence

After 48 h of infection in presence or absence of LL-37 or hBD-3, keratinocyte supernatant was removed, and the cell monolayer was washed with warm PBS (Thermofisher Scientific, Waltham, MA, USA) before being fixed with 4% paraformaldehyde (Microm Microtech, Brignais, France; Catalog No. F/P0014) for 20 min at room temperature and washed again with PBS. After removing the PBS, cells were permeabilized with 0.1% Triton X-100 in PBS for 15 min and blocked with 1% BSA in PBS for 1 h. Then keratinocytes were incubated for 1 h in presence of a viperin antibody (Merck Millipore, Darmstadt, Germany; Catalog No. MABF106) diluted to 1/50 in PBS-Tween 0.05% and then washed three times. For immunostaining, cells were incubated with a AlexaFluor 488 fluorescent secondary goat anti-mouse IgG (Invitrogen; Catalog No. 10256302) diluted to 1/200 in PBS-Tween 0.05% and cell nuclei were stained with Hoechst 33342 (Merck, Darmstadt, Germany; Catalog No. B1155) diluted to 1 µg/mL in PBS-Tween 0.05%. Fluorescent images were obtained using an IN Cell Analyzer 2200 (GE Healthcare, London, UK) with a 20× objective.

### 2.12. Type III Interferon Quantification Using HEK-Blue IFN-λ Cells

After 48 h of infection in presence or absence of LL-37 or hBD-3, keratinocyte supernatant was collected and UV inactivated at 2 joules/cm^2^ using a UV Crosslinker BIO-LINK BLX 254 (Vilber Lourmat, Collegien, France). Levels of type III IFNs were then determined, in duplicate for each sample, using IFN-λ reporter human embryonic kidney (HEK)293 cells (Invivogen; Catalog No. hkb-ifnl) in accordance with the manufacturers’ specifications.

### 2.13. Statistical Analysis

Statistical analyses were performed using the Kruskal–Wallis test followed by a Dunn’s post hoc test (GraphPad Prism 5 software). *p* values < 0.05 were considered as significant. All data were expressed as mean ± SEM from at least three independent experiments.

## 3. Results

### 3.1. LL-37 and hBD-3 Impact on Human Primary Keratinocyte Viability

The cytotoxicity of LL-37 and hBD-3 was assessed on NHEK at concentrations ranging from 1 to 20 µg/mL using a colorimetric XTT cell viability assay and an LDH assay, which measure cellular metabolic activity and plasma membrane permeabilization, respectively. After 48 h of incubation, concentrations up to 10 µg/mL had no significant impact on keratinocyte viability, which remained greater than or equal to 90%, by comparison to non-treated cells (Figure 1). Consequently, concentrations of 0.1, 1 and 10 µg/mL were chosen for the following experiments.

### 3.2. WNV Replication in Human Primary Keratinocytes Treated or Not with LL-37 and hBD-3

Human primary keratinocytes were infected for 24 h with WNV at a MOI of 0.1 in the presence or absence of LL-37 and hBD-3 at 1 or 10 µg/mL. The AMPs were added 1 h before infection and left during the 24 h replication kinetics. The viral RNA concentration was subsequently measured in the cell monolayer as well as in the cell supernatant by RT-qPCR to assess the effect of the peptides on WNV replication (Figure 2). No significant difference in viral load was observed in the cell monolayers whether or not treated with the AMPs. However, a significant 5.5-fold decrease of viral RNA was noted in the cell supernatant in the presence of 10 µg/mL of LL-37 compared with untreated cells. These results show that while LL-37 inhibits WNV replication in human primary keratinocytes, treatment with these concentrations of hBD-3 does not seem to have any effect.

### 3.3. Virucidal Properties of LL-37 and hBD-3

The virucidal properties of LL-37 and hBD-3 were then evaluated by pre-incubating for 1 h at 37 °C, a calibrated WNV inoculum of 10^5.27^ TCID_50_/mL with 1 or 10 µg/mL of LL-37 or hBD-3 before measuring the residual infectious titer on Vero cells (Figure 3). A reduction in the viral titer, compared to the control consisting of WNV incubated alone (black bars), by a factor of 14 and 95 at concentrations of 1 and 10 µg/mL, respectively, highlighted the dose-dependent direct antiviral properties of LL-37 against WNV (Figure 3A). By contrast, no significant direct antiviral effect of hBD-3 was observed at the concentrations tested (Figure 3B).

### 3.4. Immunomodulatory Properties of LL-37 and hBD-3

The effect of LL-37 and hBD-3 treatment on the immune response of the human primary keratinocytes was then evaluated. The expression of cellular inflammatory and antiviral markers such as type I (IFNβ-1) and type III (IL-28A) IFNs, IFN-stimulated genes (ISGs) (IFIT-1, OAS1, RSAD2), and chemokines (CCL-5, CXCL-8, CXCL-10) was monitored following stimulation with poly(I:C) or WNV infection in the presence of the peptides and compared with that observed after stimulation or infection without peptides, or incubation with peptides alone (Figure 4, Figure 5, Figure 6 and Figure 7).

Keratinocytes were first stimulated with 0.1 µg/mL of poly(I:C) and LL-37 or hBD-3 at final concentrations ranging from 0.1 to 10 µg/mL for 3 h (Figure 4). The results showed that mRNA expression of inflammatory markers was increased in the presence of LL-37 or hBD-3 during NHEK stimulation with poly(I:C) in comparison to cells treated with poly(I:C) alone. Indeed, treatment with 10 µg/mL of LL-37 resulted in significantly enhanced expression of CXCL-8, CCL-5 and RSAD2 mRNA with a fold increase ranging from 3.5 to 10 as compared to poly(I:C)-stimulated keratinocytes. The same pattern was observed following NHEK treatment with 1 or 10 µg/mL of hBD-3 where IFNβ-1, IL-28A, CXCL-8, CCL-5, CXCL-10, IFIT-1 and RSAD2 mRNA expression was induced from 9 to 572 times as compared to poly(I:C)-stimulated keratinocytes. Interestingly, if NHEK stimulation with poly(I:C) elicited an inflammatory response, LL-37 or hBD-3 alone exhibited no significant pro-inflammatory effects, with the notable exception of CXCL-8 mRNA expression after keratinocyte stimulation with 10 µg/mL of hBD-3 (Appendix A).

Secondly, similar experiments were performed in the context of keratinocyte infection with WNV. NHEKs were infected at an MOI of 0.1 for 24 h in the presence or absence of LL-37 and hBD-3 (Figure 5). Treatment with 10 µg/mL of hBD-3 during WNV infection resulted in a significant increase in the expression of all the ISGs and the inflammatory markers tested, except for CCL-5. A similar trend was observed for LL-37, although the fold changes were less marked and the differences were not significant.

Increased mRNA expression of RSAD2 and IFNs was then confirmed at the protein level. In the cell monolayer of keratinocytes infected in the presence or not of the peptides, the synthesis of RSAD2 also called viperin, an ISG previously reported as inhibiting flaviviral infection, was assessed by immunofluorescence assay. After 48 h of infection, immunofluorescent staining showed that the RSAD2 protein expression was enhanced in keratinocytes treated with LL-37 and hBD-3, whatever the concentration of the peptide used (Figure 6). However, this expression was significantly increased only in cells infected and treated with 1 μg/mL of each peptide.

In addition, secretion levels of IFNs-λ were measured in the cell culture supernatants using reporter HEK293 cells (Figure 7). The results showed enhanced interferon secretion in cells infected and treated with the peptides compared to infected cells alone, even if this increase was significant only in the presence of hBD-3.

All in all, the simultaneous addition of LL-37 and hBD-3 during keratinocyte stimulation by poly(I:C) or infection with WNV resulted in increased expression of the main inflammation mediators compared to poly(I:C) or virus alone. These results at the transcriptomic and proteomic levels demonstrate the immunomodulatory properties of the two peptides studied in the context of synthetic dsRNA stimulation or WNV infection.

## 4. Discussion

Antimicrobial peptides produced by skin cells represent a major weapon against infections and contribute to the maintenance of the skin microbiota. They play a major role in the barrier function exerted by the skin, through direct antimicrobial activities against pathogens or indirect antimicrobial activities by stimulating innate immunity. If their antibacterial and antifungal roles are now well-known, their antiviral properties remain little studied. Among the nine peptides known to be secreted by the human keratinocyte, our interest focused on LL-37 and hBD-3, whose secretion is induced during the infection of skin cells by flaviviruses [29,38,39]. This research aimed to assess the effect of these AMPs during infection of human primary keratinocytes with WNV, an emerging flavivirus in Europe, inoculated within the skin during the blood meal of the mosquito vector.

We first studied the antiviral properties of LL-37 and hBD-3 during infection kinetics of human primary keratinocytes. A reduction in the viral RNA concentration in the culture supernatant of keratinocytes treated with LL-37 was observed. In addition, a 2 log_10_-reduction in the viral infectious titer of a WNV strain incubated with LL-37 was also shown, suggesting direct antiviral activity against WNV. This activity may be linked to an alteration of the viral particle, and more precisely of its envelope, by the antimicrobial peptide. This is a classic mode of action of antimicrobial peptides related to their cationic and amphiphilic nature, which allows them to create pores within the viral envelope [16,17,40,41]. The ability of LL-37 to degrade the viral envelope has been demonstrated against a large panel of viruses. For example, electronic microscopy has revealed the disruption of the vaccinia virus envelope or influenza A virus exposed to LL-37 [16,17]. LL-37-related inhibition of human herpes virus 8 internalization in oral epithelial cells relies on the same mechanism [34]. Finally, pre-incubation of the arboviruses DENV, ZIKV and Venezuelan encephalitis equine virus (VEEV) with LL-37, or one of its analogs, has resulted in a significant decrease in the number of active virions suggesting, here again, an alteration of the viral particle [22,33,35,42,43]. In the previous studies, degradation of the viral envelope was observed from an LL-37 concentration of 10 μg/mL, therefore, identical to the highest concentration tested during this work [15,34]. LL-37 has also been shown to exert direct antiviral properties by inhibiting the endosomal cell entry of Ebola virus or by inhibiting enterovirus A71 cell attachment [44,45]. Otherwise, the immunomodulatory properties of LL-37 observed during WNV infection, especially stimulation of the expression of several ISGs, are also likely to contribute to the reduced viral replication observed in treated keratinocytes. In particular, we noted increased expression of RSAD2 in keratinocytes infected with WNV in the presence of LL-37. RSAD2, also called viperin, is a well-known antiflavivirus ISG [46,47,48,49,50]. It converts cytidine triphosphate into 3′-deoxy-3′,4′-didehydro-CTP, which acts as a chain terminator for the flavivirus RNA-dependent RNA-polymerase in mammalian cells [51]. RSAD2 also inhibits several steps of viral translation, as demonstrated during HEK 293 cell infection with WNV or DENV [52]. RSAD2 binds and leads to the proteasomal degradation of flaviviral NS3 protease [53]. Finally, RSAD2 blocks viral budding, which could explain the reduction of WNV RNA in keratinocyte supernatant, whereas RNA concentration is increased within the cell monolayer (reviewed in [46]).

Conversely, our results suggest an absence of direct antiviral activity of hBD-3 against WNV at the concentrations tested. Indeed, the viral load in the keratinocyte monolayer as well as in the supernatant was not reduced by the addition of the peptide. Likewise, incubation of the virus with hBD-3 did not reduce the infectious titer of WNV. These data are consistent with previous work showing that a pretreatment by 30 µg/mL of hBD-3 did not result in a significant decrease in DENV titer, contrary to that observed with LL-37 [28]. Nevertheless, other studies have shown that this peptide can bind the envelope glycoprotein B of herpes simplex virus or its host cell receptors, thereby preventing viral entry [32]. The ability of hBD-3 to bind human immunodeficiency virus particles and to downmodulate its cell surface receptor CXCR4 has also been demonstrated [54]. This antiviral effect is concentration dependent and significant for hBD-3 concentrations greater than 25 µg/mL, which are cytotoxic for human primary keratinocytes. On the other hand, we showed that hBD-3 can stimulate keratinocyte innate immune response during poly(I:C) stimulation or WNV infection. Some pro-inflammatory properties of hBD-3 have been previously described. hBD-3 can induce inflammatory response by itself, through CCR6 binding or the G protein-coupled receptor (GPCR) and phospholipase C (PLC) signaling pathway, leading to the release of several cytokines such as IL-6, IL-10, IL-37, IFN-γ, CXL-10, CCL-2 and CCL-5 by human keratinocytes [55,56]. It can also modulate the response to TLR agonists. Semple et al. observed increased production of IFN-β, TNF-α, CXCL-8 and IL-6 in monocytes and PBMCs stimulated with poly(I:C) in the presence of hBD-3 [57]. Higher concentrations of IFN-β and TNF-α were observed in the sera of transgenic mice expressing hBD-3 and stimulated with poly(I:C) as compared to control mice [57]. Our study is, however, the first description of the potentiating effect of hBD-3 on the inflammatory response of the keratinocyte in the context of viral infection, highlighting the ability of this AMP to boost the antiviral response of the skin through the induction of IFNs and ISGs. On the other hand, LL-37 has also shown immunomodulatory properties. It has previously been shown to increase pro-inflammatory cytokine expression such as IFNβ-1, IL-6 and CXCL-8 induced by poly(I:C) stimulation in human keratinocytes [58,59]. LL-37 also modulates inflammatory response induced by viral infection. The addition of LL-37 to human rhinovirus-infected human bronchial epithelial cells enhances IL-6 and CCL-2 production while increasing the expression of type I IFN during VEEV infection [42,58]. This enhancement of the inflammatory response requires TLR3, which is activated by the complex formed by the association between LL-37 and dsRNAs [58]. Interestingly, this mechanism was demonstrated to be cell-specific and concentration-dependent since high concentrations of LL-37, from 25 µg/mL, were associated with decreased pro-inflammatory cytokine release in the cell supernatant [60]. The secretion by infected cells of inflammatory mediators, especially IFNs, represents an important antiviral weapon not only for the cell itself, but also for the neighboring ones, as this mediator can exert both auto- and paracrine effects. This phenomenon could consequently prepare the non-infected surrounding cells for a potential viral infection by inducing the synthesis of antiviral mediators as ISGs. Taken together, our results suggest that both LL-37 and hBD-3 secreted within the skin can interplay a role indefense against the early steps of WNV infection. Further studies should evaluate the immunomodulatory properties of hBD-3 in other cell types, whether after stimulation with poly(I:C) or viral infection.

## 5. Conclusions

Our results confirmed the major immunostimulatory role of LL-37 and hBD-3 through induction of the production of cytokines and chemokines in NHEK in combination with the ability of LL-37 to directly inactivate the enveloped WNV. This study highlights the antiviral potentialities of these two AMPs produced directly at the site of viral inoculation by the mosquito and the site of viral replication, thereby suggesting their role in the pathophysiology of the early steps of infection. The characterization of such activities could help to devise new antiviral strategies, through the design of novel therapeutic analogs against infection or the evaluation of agonists capable of specifically stimulating antiviral response.

## Figures and Tables

**Figure 1 viruses-14-01552-f001:**
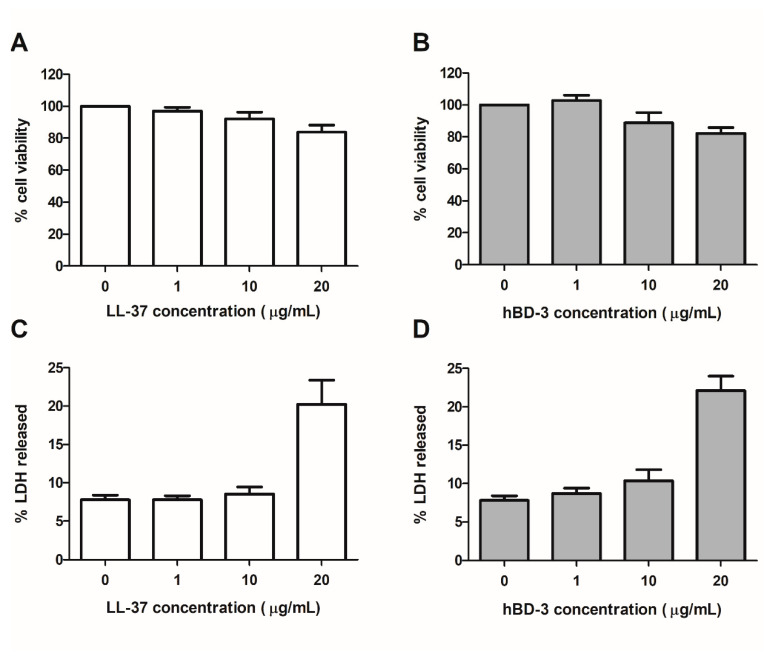
Cytotoxicity assays. The cytotoxicity of concentrations of LL-37 or hBD-3 ranging from 1 to 20 µg/mL was assessed on primary human keratinocytes after 48 h of incubation using XTT tests (**A**,**B**) or lactate dehydrogenase (LDH) release assay (**C**,**D**). Mean ± SEM of at least 3 independent experiments are presented here.

**Figure 2 viruses-14-01552-f002:**
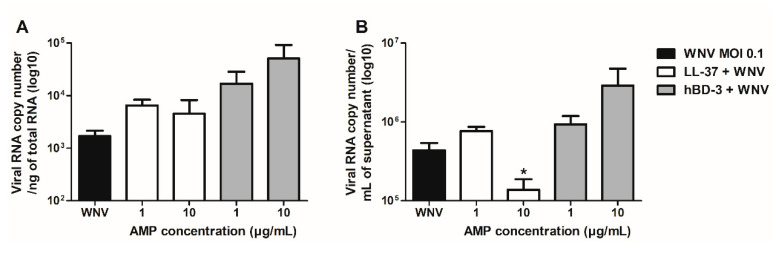
WNV RNA quantification in the cell monolayer (**A**) and supernatant (**B**) during NHEK infection by WNV at a MOI of 0.1 in presence or not of 1 or 10 µg/mL of LL-37 or hBD-3 for 24 h. Mean +/− SEM of 5 independent experiments are presented here. * *p* < 0.05.

**Figure 3 viruses-14-01552-f003:**
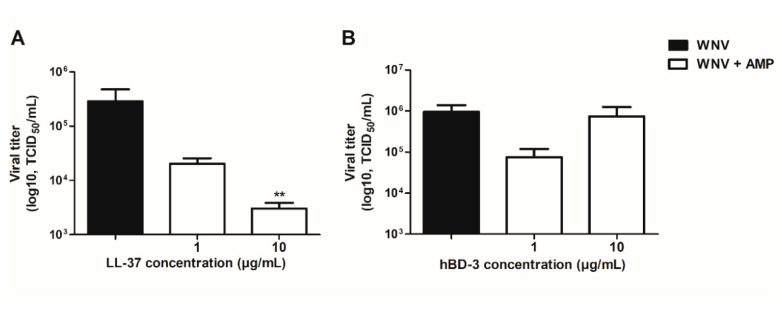
A WNV inoculum of 10^5.27^ TCID_50_/mL was incubated for 2 h at 37 °C without or with 1 and 10 µg/mL of LL-37 (**A**) or hBD-3 (**B**). The residual infectious titer was then measured on Vero cells. Mean ± SEM of, respectively, 5 and 4 independent experiments are presented here. ** *p* < 0.01.

**Figure 4 viruses-14-01552-f004:**
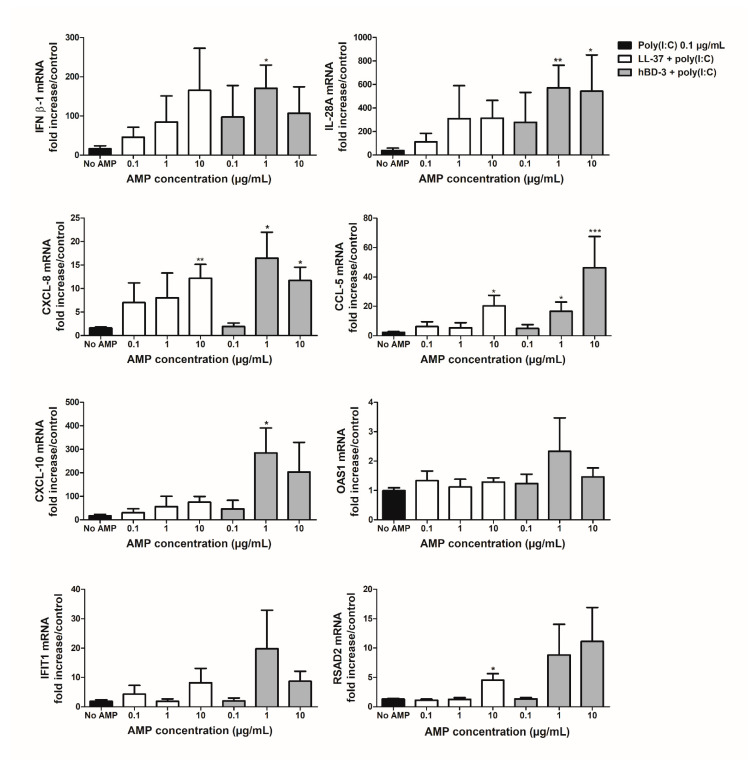
Inflammatory response of NHEK stimulated with 0.1 µg/mL of poly(I:C) alone or in combination with increasing concentrations of LL-37 or hBD-3 for 3 h. Results are expressed as the fold increase above unstimulated keratinocytes. Mean ± SEM of 6 independent experiments are presented here. * *p* < 0.05, ** *p* < 0.01 and *** *p* < 0.001.

**Figure 5 viruses-14-01552-f005:**
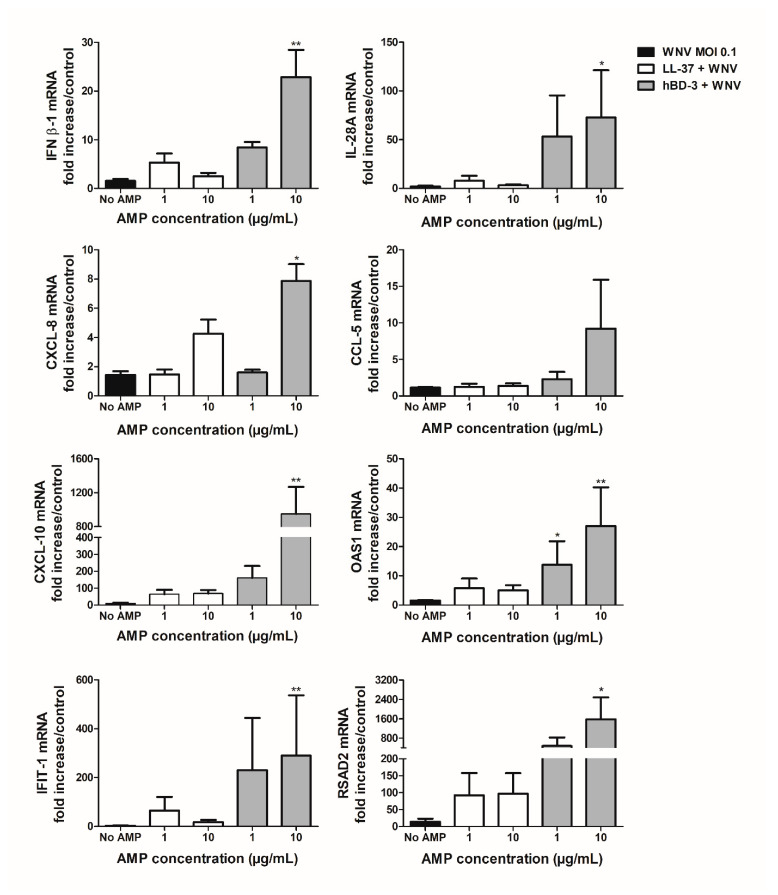
Inflammatory response of NHEK infected by WNV at a MOI of 0.1 in presence or not of 1 or 10 µg/mL of LL-37 or hBD-3 for 24 h. Results are expressed as the fold increase above uninfected keratinocytes. Mean ± SEM of 5 independent experiments are presented here. * *p* < 0.05 and ** *p* < 0.01.

**Figure 6 viruses-14-01552-f006:**
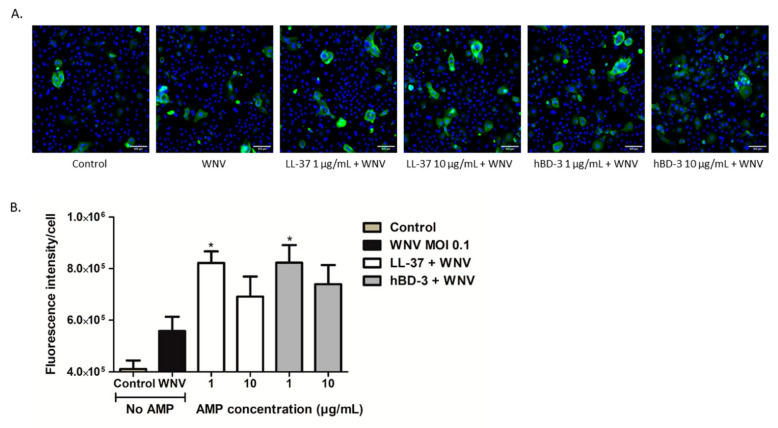
Immunostaining of viperin (RSAD2) in human primary keratinocytes infected with WNV in presence or not of LL-37 and hBD-3. Keratinocytes were infected by WNV at a MOI of 0.1 and stimulated with 1 or 10 µg/mL of LL-37 or hBD-3 for 48 h. RSAD2 was quantified in each cell by immunofluorescence. A 1/100 diluted viperin antibody was first added for one hour, and then cells were incubated with a fluorescent secondary goat anti-mouse IgG; cell nuclei were stained with Hoechst 33342. (**A**) Fluorescent images were obtained using an IN Cell Analyzer 2200 (GE Healthcare, London, UK) with a 20× objective (scale bar: 100 µm). (**B**) Fluorescence intensity per cell was calculated by taking the average of 5 pictures per well and 6 wells per condition, and relating it to the number of cell nuclei per picture. Mean ± SEM of 3 independent experiments are presented here. Statistical analysis has been performed compared to WNV alone. * *p* < 0.05.

**Figure 7 viruses-14-01552-f007:**
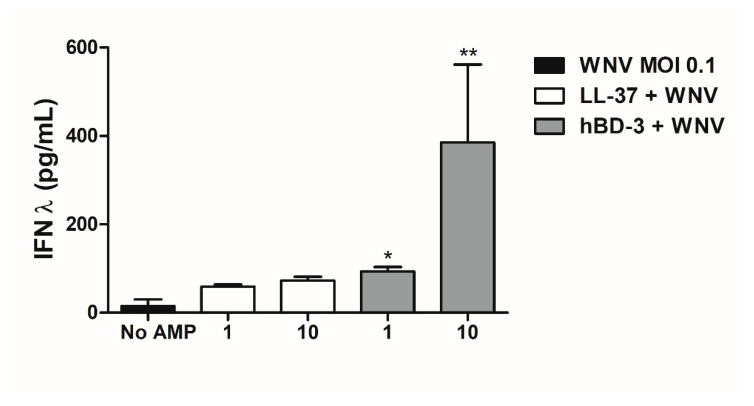
IFN-λ secretion determination using HEK-blue cells in supernatant of human primary keratinocytes infected by WNV at a MOI of 0.1 and stimulated with 1 or 10 µg/mL of LL-37 or h-BD3 for 48 h. Results are presented as mean ± SEM of 4 independent experiments. * *p* < 0.05 and ** *p* < 0.01.

## Data Availability

Data contained within the article or Appendix A are available on request from the authors.

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
