# Peer review of "Antiviral Effect of hBD-3 and LL-37 during Human Primary Keratinocyte Infection with West Nile Virus"

_viruses, 2022, doi:10.3390/v14071552_

Round 1

Reviewer 1 Report

Dear Authors,

It was a pleasure to read your manuscript. Even though similar studies have been done with other viruses, your results are interesting and give a new perspective on the innate response to WNV infection.  Overall, a few minor issues should be addressed.

The Material and methods should be clear about how many replicates for each experiment were done. 

Line 423. 9 should be nine

Line 439. The full name of HHV-8 should be provided.

Author Response

Responses to reviewer 1 comments:

Dear Authors,

It was a pleasure to read your manuscript. Even though similar studies have been done with other viruses, your results are interesting and give a new perspective on the innate response to WNV infection.  Overall, a few minor issues should be addressed.

The Material and methods should be clear about how many replicates for each experiment were done. 

Line 423. 9 should be nine

It has been corrected.

Line 439. The full name of HHV-8 should be provided.

It has been precised.

Reviewer 2 Report

Manuscript Review: viruses-1754195-peer-review-v1

Antiviral effect of hBD-3 and LL-37 during human primary keratinocyte infection with West Nile virus

Céline Chessa, Charles Bodet, Clément Jousselin, Andy Larivière, Alexia Damour, Julien Garnier, Nicolas Lévêque, Magali Garcia

Summary:

Chessa, et al. present a manuscript that aims to reveal the antiviral role of two human antimicrobial peptides (LL-37 and hBD-3) against WNV infection in a highly relevant model of the epidermal site of infection. The authors demonstrate that LL-37 has a direct role in inactivation of infectious WNV virions, and both peptides can modulate the innate immune responses of keratinocytes in the context of active PRR engagement. This article presents an important field of work that is sorely neglected and should thus be well received by the research community. This manuscript shows great promise, however there are a few aspects that require attention before it is suitable for publication.

Comments:

1.       There are a few notable details missing from the Materials and Methods that are essential for both the interpretation of the data, and any attempt to reproduce results in other laboratories.

·         The catalogue numbers are missing from many reagents, most notably the LL-37 and hBD-3 peptides, the anti-viperin antibody, the fluorescently-tagged goat anti-mouse IgG, and the IFN-λ reporter HEK293 cells (various polymerases and kits could also benefit from listing the catalogue numbers).

·         The identity of the conjugate for the fluorescently-tagged goat anti-mouse IgG is not reported, and neither is the concentration used, nor the buffer it is diluted in.

·         The concentration and buffer conditions are similarly not listed for the Hoechst 33342.

·         The buffer conditions are not listed for incubation with the primary anti-viperin antibody.

·         Some details of the poly(I:C) treatment and WNV infection are lacking. Was the poly(I:C) added to the cell culture media (i.e. triggering TLR3), or was it transfected (triggering RIG-I/MDA5) – neither option is clearly stated. For the WNV infection, how was the inoculum applied? Was it incubated in a minimal volume for 2hrs at 37oC with gentle rocking, then removed before culture media was replaced, or was the virus simply added to the cell culture medium and the inoculum left in place for the entirety of the experiment? This may greatly affect interpretation of the results. In the former option, the antimicrobial peptides used to treat cells prior to infection may be removed along with the virus inoculum (as opposed to remaining present in the culture media and so potentially stimulating cells over a longer period). In the latter option, some of the virion RNA present in the inoculum may still be present in the supernatant measured in Figure 2B without ever entering a cell or replicating (thus the values may not truly represent productive replication leading to new particle formation and secretion).

·         The acronyms FBS and SVF are both used, yet a definition is given for neither. I would suggest that FBS is the more familiar term for readers, yet should still be defined.

·         In line 193, the amount of virus is given as 1.105.3 TCID50. If the first point is a decimal, the value is 1.657224 TCID50 whereas if the notation is intended to be 1x105.3 TCID50 the value is 199,526.231 TCID50. I assume the latter was intended, and suggest the 1x105.3 TCID50 notation is used.

·         Regarding the statistics used, I would suggest that the Mann Whitney test is inappropriate for use if the intention is to perform several of these tests comparing 2 groups at a time. The statistics for the data presented is most suitably measured using a multiple comparisons test such as Kruskal-Wallis or ANOVA, followed by a post-test. Each figure deals with multiple samples and variables (dose, identity of AMP, etc) that are all compared together in the one assay. These are multiple comparisons.

2.       Figure 2: There is no P-value listed for the significantly different sample, and the figure legend does not describe the meaning of the asterisk.

3.       Figure 3: Could the authors confirm that the WNV control (black bars) was also incubated for 2 hours at 37oC? It is not specifically stated. Was this control incubated with vehicle (i.e. the diluent for the AMPs) at that time?

4.       Figure 4: This data set would benefit from the inclusion of an unstimulated control sample to demonstrate that these primary keratinocytes respond to poly(I:C) stimulation normally and induce each of these genes as expected. This would also demonstrate how robust or poor this level of expression is in the absence of assistance from the AMPs.

5.       Figure 5: As for the comment for Figure 4, this data set would benefit from inclusion of uninfected controls. Additionally, it is worth considering that unlike poly(I:C), WNV abundance within the keratinocytes (and thus the abundance of PRR ligand available to stimulate responses) is directly affected by the AMPs (cross-reference with Figure 2). If the total RNA harvested for this data is the same as that used in Figure 2A (i.e. they are matched per experiment), I would suggest that it may be beneficial to normalize the innate immune gene values to the WNV genome copy numbers (may be suitable as a supplementary figure).

6.       Figure 6: This figure should be accompanied by representative images of viperin abundance. It would be beneficial to demonstrate whether increased viperin abundance is uniform within the cell population, or if there are discrete cells with greater abundance than neighbours (even better than this would be co-staining against WNV to determine if it is directly infected cells or bystanders that have greater or lesser virperin production – however I appreciate such new experimental data may move beyond the scope of this report). Furthermore, it is important to provide further information regarding what cells were chosen to quantify, and how many were quantified per independent experiment. Was quantification performed on all cells with a field and averaged? Were a particular number of cells selected at random for quantification? Was another method used? Lastly, it would certainly be beneficial to include an immunoblot data set of matched samples. Immunoblots can clearly demonstrate protein abundance, and do not suffer from many of the variables that affect imaging (such as of some cells within a population aligning slightly out of the focal plane, leading to inaccurate quantification of signal intensity).

Author Response

Responses to reviewer 2 comments:

There are a few notable details missing from the Materials and Methods that are essential for both the interpretation of the data, and any attempt to reproduce results in other laboratories.

The catalogue numbers are missing from many reagents, most notably the LL-37 and hBD-3 peptides, the anti-viperin antibody, the fluorescently-tagged goat anti-mouse IgG, and the IFN-λ reporter HEK293 cells (various polymerases and kits could also benefit from listing the catalogue numbers).The identity of the conjugate for the fluorescently-tagged goat anti-mouse IgG is not reported, and neither is the concentration used, nor the buffer it is diluted in. The concentration and buffer conditions are similarly not listed for the Hoechst 33342. The buffer conditions are not listed for incubation with the primary anti-viperin antibody.

      Catalogue numbers were indicated as requested and missing information regarding concentrations and buffers has been added to the text.

Some details of the poly(I:C) treatment and WNV infection are lacking. Was the poly(I:C) added to the cell culture media (i.e. triggering TLR3), or was it transfected (triggering RIG-I/MDA5) – neither option is clearly stated.

Poly(I:C) was added to the cell culture media. This point was also clarified in the text.

For the WNV infection, how was the inoculum applied? Was it incubated in a minimal volume for 2hrs at 37oC with gentle rocking, then removed before culture media was replaced, or was the virus simply added to the cell culture medium and the inoculum left in place for the entirety of the experiment? This may greatly affect interpretation of the results. In the former option, the antimicrobial peptides used to treat cells prior to infection may be removed along with the virus inoculum (as opposed to remaining present in the culture media and so potentially stimulating cells over a longer period). In the latter option, some of the virion RNA present in the inoculum may still be present in the supernatant measured in Figure 2B without ever entering a cell or replicating (thus the values may not truly represent productive replication leading to new particle formation and secretion).

The peptide was brought into contact with the cells at the final concentration of 1 and 10 µg/mL in 1 mL of culture medium for one hour before infection. The virus was then added to the culture medium containing the peptide at a concentration giving an MOI of 0.1. Virus and peptide were left in contact with the cells for 24 hours (virus replication) and 48 h (inflammatory marker expression). The viral load in the cell monolayer and the cell culture supernatant was measured by RT-qPCR at the end of the 24 h incubation and compared between treated and not treated cells. After 48 h incubation, cell monolayers and cell culture supernatants were used to monitor inflammatory marker expression at the protein level. These clarifications have been added to the Materials and Methods.

The acronyms FBS and SVF are both used, yet a definition is given for neither. I would suggest that FBS is the more familiar term for readers, yet should still be defined.

That’s true, it was incorrect, SVF has been replaced by FBS lines 184 and 185.

In line 193, the amount of virus is given as 1.105.3 TCID50. If the first point is a decimal, the value is 1.657224 TCID50 whereas if the notation is intended to be 1x105.3 TCID50 the value is 199,526.231 TCID50. I assume the latter was intended, and suggest the 1x105.3 TCID50 notation is used.

This has been changed according to reviewer’s recommendations.

Regarding the statistics used, I would suggest that the Mann Whitney test is inappropriate for use if the intention is to perform several of these tests comparing 2 groups at a time. The statistics for the data presented is most suitably measured using a multiple comparisons test such as Kruskal-Wallis or ANOVA, followed by a post-test. Each figure deals with multiple samples and variables (dose, identity of AMP, etc) that are all compared together in the one assay. These are multiple comparisons.

All the statistical analyses have been redone performing a Kruskall Wallis test followed by a Dunn’s post-test. This has been specified in the material and method and the figures have been modified when changes were observed.

  1. Figure 2: There is no P-value listed for the significantly different sample, and the figure legend does not describe the meaning of the asterisk.

The p value has been added.

  1. Figure 3: Could the authors confirm that the WNV control (black bars) was also incubated for 2 hours at 37oC? It is not specifically stated. Was this control incubated with vehicle (i.e. the diluent for the AMPs) at that time?

We confirm that WNV control was incubated with the cells under the same conditions as those used for the incubation in the presence of the peptides (for 2 hours at 37°C). The diluent for the AMPs was not added to the virus during this experiment since water was the buffer used to reconstitute the peptides.

  1. Figure 4: This data set would benefit from the inclusion of an unstimulated control sample to demonstrate that these primary keratinocytes respond to poly(I:C) stimulation normally and induce each of these genes as expected. This would also demonstrate how robust or poor this level of expression is in the absence of assistance from the AMPs.

In the figure 4, gene expression levels after stimulation with poly(I:C) in the absence (black bars) or in the presence of LL-37 (white bars) and hBD-3 (grey bars) are already expressed by comparison to a control corresponding to unstimulated human primary keratinocytes. What is shown in the graph are fold increase above unstimulated cells. The sentence: “Results are expressed as the fold increase above unstimulated keratinocytes.” Was added to the legend of the figures.

  1. Figure 5: As for the comment for Figure 4, this data set would benefit from inclusion of uninfected controls. Additionally, it is worth considering that unlike poly(I:C), WNV abundance within the keratinocytes (and thus the abundance of PRR ligand available to stimulate responses) is directly affected by the AMPs (cross-reference with Figure 2). If the total RNA harvested for this data is the same as that used in Figure 2A (i.e. they are matched per experiment), I would suggest that it may be beneficial to normalize the innate immune gene values to the WNV genome copy numbers (may be suitable as a supplementary figure).

In the same way as for Figure 4, the results in Figure 5 are expressed as a fold increase relative to the uninfected and unstimulated keratinocytes used as reference. This was clarified in the figure legend. Indeed, the study of the inflammatory response of keratinocytes to WNV infection in the presence or absence of peptides was carried out using the same RNA extracts as those used for the quantification of the WNV viral load in the monolayer presented in figure 2. Due to the immunomodolutary effect of the peptides observed in some conditions, the results didn’t always show a correlation between viral replication and the expression of inflammatory markers.

We think it's less confusing to present the data this way.

  1. Figure 6: This figure should be accompanied by representative images of viperin abundance.

The images of the immunofluorescence assay have been added to the figure 6 (figure 6A).

It would be beneficial to demonstrate whether increased viperin abundance is uniform within the cell population, or if there are discrete cells with greater abundance than neighbours (even better than this would be co-staining against WNV to determine if it is directly infected cells or bystanders that have greater or lesser virperin production – however I appreciate such new experimental data may move beyond the scope of this report).

Furthermore, it is important to provide further information regarding what cells were chosen to quantify, and how many were quantified per independent experiment. Was quantification performed on all cells with a field and averaged? Were a particular number of cells selected at random for quantification? Was another method used?

Fluorescence intensity per cell was calculated by taking the average of 5 pictures per well and 6 wells per condition, and then relating it to the number of cell nuclei per picture, as now specified in the manuscript.

Lastly, it would certainly be beneficial to include an immunoblot data set of matched samples. Immunoblots can clearly demonstrate protein abundance, and do not suffer from many of the variables that affect imaging (such as of some cells within a population aligning slightly out of the focal plane, leading to inaccurate quantification of signal intensity).

We did not perform an immunoblot analysis considering that the data obtained by immunofluorescence were consistent with those of the RT-qPCR analysis for this same protein but also with the IFN-λ assays. It therefore did not seem necessary to us to provide additional proof of the over-expression of inflammation markers induced by antimicrobial peptides during WNV infection.